# Laboratory Comparison of Low-Cost Particulate Matter Sensors to Measure Transient Events of Pollution—Part B—Particle Number Concentrations

**DOI:** 10.3390/s23177657

**Published:** 2023-09-04

**Authors:** Florentin Michel Jacques Bulot, Hugo Savill Russell, Mohsen Rezaei, Matthew Stanley Johnson, Steven James Ossont, Andrew Kevin Richard Morris, Philip James Basford, Natasha Hazel Celeste Easton, Hazel Louise Mitchell, Gavin Lee Foster, Matthew Loxham, Simon James Cox

**Affiliations:** 1Faculty of Engineering and Physical Sciences, University of Southampton, Southampton SO17 1BJ, UK; p.j.basford@soton.ac.uk (P.J.B.); hlm1g16@soton.ac.uk (H.L.M.); s.j.cox@soton.ac.uk (S.J.C.); 2Southampton Marine and Maritime Institute, University of Southampton, Southampton SO16 7QF, UK; nhcs1g13@soton.ac.uk (N.H.C.E.); m.loxham@soton.ac.uk (M.L.); 3Danish Big Data Centre for Environment and Health (BERTHA), Aarhus University, DK-4000 Roskilde, Denmark; hugo.russell@envs.au.dk; 4AirScape UK, London W1U 6TQ, UK; matthew.johnson@airscape.ai; 5Department of Environmental Science, Atmospheric Measurement, Aarhus University, DK-4000 Roskilde, Denmark; 6Department of Chemistry, University of Copenhagen, DK-2100 Copenhagen, Denmark; mohsen@chem.ku.dk; 7BizData, Melbourne, VIC 3000, Australia; steven.ossont@bizdata.co.nz; 8National Oceanography Centre, Southampton SO14 3ZH, UK; andmor@noc.ac.uk; 9School of Ocean and Earth Science, National Oceanography Centre, University of Southampton, Southampton SO14 3ZH, UK; gavin.foster@noc.soton.ac.uk; 10Faculty of Medicine, University of Southampton, Southampton SO17 1BJ, UK; 11National Institute for Health Research, Southampton Biomedical Research Centre, Southampton SO16 6YD, UK; 12Institute for Life Sciences, University of Southampton, Southampton SO17 1BJ, UK

**Keywords:** low-cost sensors, particle number concentration, laboratory study, fine particles, particulate matter, air pollution

## Abstract

Low-cost Particulate Matter (PM) sensors offer an excellent opportunity to improve our knowledge about this type of pollution. Their size and cost, which support multi-node network deployment, along with their temporal resolution, enable them to report fine spatio-temporal resolution for a given area. These sensors have known issues across performance metrics. Generally, the literature focuses on the PM mass concentration reported by these sensors, but some models of sensors also report Particle Number Concentrations (PNCs) segregated into different PM size ranges. In this study, eight units each of Alphasense OPC-R1, Plantower PMS5003 and Sensirion SPS30 have been exposed, under controlled conditions, to short-lived peaks of PM generated using two different combustion sources of PM, exposing the sensors’ to different particle size distributions to quantify and better understand the low-cost sensors performance across a range of relevant environmental ranges. The PNCs reported by the sensors were analysed to characterise sensor-reported particle size distribution, to determine whether sensor-reported PNCs can follow the transient variations of PM observed by the reference instruments and to determine the relative impact of different variables on the performances of the sensors. This study shows that the Alphasense OPC-R1 reported at least five size ranges independently from each other, that the Sensirion SPS30 reported two size ranges independently from each other and that all the size ranges reported by the Plantower PMS5003 were not independent of each other. It demonstrates that all sensors tested here could track the fine temporal variation of PNCs, that the Alphasense OPC-R1 could closely follow the variations of size distribution between the two sources of PM, and it shows that particle size distribution and composition are more impactful on sensor measurements than relative humidity.

## 1. Introduction

Exposure to air pollution is a major cause of environmental morbidity and mortality in the world at present, with Particulate Matter (PM) air pollution being associated with 8.9 million premature deaths per year [1,2]. PM air pollution varies with fine spatio-temporal granularity and can have heterogeneous composition and concentration over a specific area [3]. Current regulatory monitoring networks are based on cumbersome and expensive apparatus that means monitoring with the spatial coverage required to comprehensively understand the spread of air pollution is not feasible. Given the recently substantially reduced WHO exposure limits [4], down to 5 μg/m${}^{3}$ as an annual mean for PM${}_{2.5}$, there is an increased need for monitoring. At this lower threshold, local sources can often be the factor causing exceedance, which makes information concerning local levels and sources more important than they have been in the past [5]. The EU is moving towards adopting the more stringent WHO standard [6] and voices in the community are saying that the only way to ensure compliance is by using dense networks of low-cost sensors in populated areas [7].

Low-cost PM sensors have been used in the literature and in various projects around the world to determine PM mass concentrations, especially when deployed as networks of sensors to improve the limited spatio-temporal coverage of existing monitoring networks [5]. Considerable research has been conducted to reach a known level of precision and accuracy with some studies achieving the data quality objectives of reference-grade instruments with the proper calibration methods and frequencies [8], at high temporal resolution, providing data that was not previously available to determine population exposure to PM air pollution at a finer level. However, some of these sensors provide not only PM concentrations but also Particle Number Concentrations (PNCs) for different size ranges, for example by giving PNC in the range 0.3–1 μm and PNC in the range 1–2.5 μm. They are based on light scattering and generally claim to measure particles of diameters 0.3–10 μm, and it is important to note that the scattering efficiency decreases as the diameter is close to or lower than 0.3 μm therefore implying a lower performance in the response to the lower size ranges of particles. Standard metrics for PM have evolved through the last few decades [9]. For example, in the US, the 1971 National Ambient Air Quality Standards was set for PM as total suspended particles. Later in 1987, following new evidence on the health effects of PM, the standards were revised to focus on PM_10_. In 1997, the first standards for PM_2.5_ were issued to account for the health impact of this size fraction. Although current legal limits are based on PM mass concentration, not all PM is equally harmful and other properties of the particles may be significant in terms of health impact, such as their composition, shape, size, etc. [10]. Therefore, size distribution of PM could be a promising metrics to better capture the health impact of PM.

There is variation in what is reported with some sensors giving a detailed size distribution and others only outputting PNC with a restricted number of size ranges. For instance, the Plantower PMS5003 outputs six different size ranges and the Alphasense OPC-R1 outputs 13 size ranges. The ability of the sensors to report PNCs of different size fractions can be used to identify sources of pollution. Indeed, in Delhi, India, Hagan et al. [11] used the first three size ranges of an Alphasense OPC-N2, in conjunction with data on other air pollutants (CO, NO${}_{2}$, SO${}_{2}$, O${}_{3}$) to successfully identify sources of pollution using positive matrix factorisation. Additionally, we previously demonstrated reference-grade improvements to the performances of Plantower PMS5003 and Sensirion SPS30 through calibration methods based on the PNCs reported by these sensors [12]. Similarly, Wallace et al. [13,14] developed a calibration method using the PNCs reported by the Plantower PMS5003 that outperformed calibration methods based on mass concentrations. This improvement in performances was confirmed by further long-term studies [15,16].

There are broadly two types of low-cost PM sensors [17]: (1) volume scattering, or integrating nephelometers, that measure the light scattered by an ensemble of particles; and (2) single particle counters which count individual particles. The two types have different sensitivities to aerosol parameters and environmental factors [18]. However, there is disagreement in the literature about which sensor belongs to which type. There is also concern about whether these low-cost PM sensors can accurately segregate PNCs into different size ranges [17,18,19,20,21]. Recently, Ouimette et al. [22] conducted a detailed study of the PurpleAir sensors (PurpleAir, Draper, UT, USA) (which use two Plantower PMS5003s (Plantower, Nanchang City, China)) comparing them to a research-grade integrating nephelometer and developed a physical model that showed that the Plantower PMS5003 is a cell-reciprocal nephelometer providing a reliable measurement of the aerosol scattering coefficients for particles in the range 0.26–0.46 μm. Ouimette et al. [22] is one of the rare studies that focused on sensor-reported PNCs. A few laboratory studies have been conducted regarding the size segregation capacity of the sensors [19,21,23,24]. Three of these studies have focused only on sensor-reported mass concentrations, while one has also studied sensor-reported PNCs. All of the above studies examined sensor performances with stable concentrations of PM over periods ranging from 5 min to 1 h, depending on the study. They exposed the sensors to PM of a variety of sources and sizes. Several studies highlighted that low-cost PM sensors are susceptible to a range of environmental factors, namely particle composition, size distribution and Relative Humidity (RH). However, different studies obtained contrasting results concerning RH which suggests that other factors may be at play that are not accounted for.

Feature selection methods quantify the contribution of individual features (here environmental factors) to the variability of an output variable (here sensor-reported PNCs) [25]. They are one of the most popular techniques to improve the explainability of machine learning models [26], which are often used to correct measurements from low-cost sensors. Feature selection methods are divided into three sub-categories of method: filter-based, wrapper-based and embedded methods [27]. Filter-based methods class the variables using different metrics such as the Pearson coefficient or the Akaike information criterion (AIC). They do not account for possible correlation between variables and are prone to missing patterns [28]. Wrapper-based methods iteratively use supervised learning techniques (e.g., linear model, support vector machine) to classify the variables. They apply algorithms such as recursive feature selection and greedy forward selection. They are generally more accurate than filter-based methods but risk over-fitting and are more computationally intensive [28]. Finally, embedded methods have the reduction of the number of variables embedded in their algorithms, such as Lasso regression, elastic net regression or random forest. They constitute a trade-off between filter and wrapper-based methods [29]. Nonetheless, the features selected are dependent on the methods chosen and the best practice is to use different methods concomitantly and to compare their results [25].

The current study is the second part of a comprehensive experiment that aimed to characterise the response of a range of low-cost PM sensors to transient events of PM pollution. The first part of this study [30] focused on sensor-reported mass concentrations while this current contribution focuses on sensor-reported PNCs. Sensors measuring at a 10 s temporal resolution were exposed to short-lived peaks of PM pollution (≈1 min) generated by lighting candles and incense sticks at different RH levels. Using two combustion sources, we can assess the performance of the sensors across different size distributions. Understanding the response of these sensors to short-lived events of PM pollution is important especially if these sensors are to be used indoors, where polluting activities may last only for a few minutes [31], or used outdoors as a network for tracking events of PM pollution as they spread through an area [32]. These data could also be integrated into models to further their granularity through data fusion techniques [33]. Here, we compare eight units of each sensor model, Sensirion SPS30, Plantower PMS5003 and Alphasense OPC-R1, at a total of 24 low-cost PM sensors, at 10 s resolution. A TSI OPS 3330 (TSI Inc., Shoreview, MN, USA)is used as a reference instrument. The aim is to characterise sensor-reported particle size distribution, to determine whether sensor-reported PNCs can follow the transient variations of PM observed by the reference instruments and to determine the relative impact of different variables on the performances of the sensors.

The objectives of this study are:To determine whether the sensors can be used at a high temporal resolution to follow trends of PNCs.To determine whether the different sensor-reported PNCs are independent from each other and characterise their accuracy.To characterise the capacity of these sensors to capture the size distribution of PM.To determine environmental factors impacting the response of the sensors.

## 2. Materials and Methods

### 2.1. Low-Cost PM Sensors

The low-cost sensors were mounted in the air quality monitors developed in Johnston et al. [34], without their environmental enclosure as can be seen in Figure 1. The absence of enclosure helps reduce potential residual heat build-up in the vicinity of the sensors. The low-cost sensors studied here are the Plantower PMS5003, the Sensirion SPS30 and the Alphasense OPC-R1. These sensors were chosen because they output PNCs of the PM measured for different size fractions. Table 1 presents the different size ranges of these three models of sensors. The Honeywell HPMA115S0 and the Novafitness SDS018 were also measuring during the experiment but the data they produced were not used in this study as they only report PM mass concentrations. All the low-cost sensors tested here are optical measurement devices based on Mie light-scattering. Four air quality monitors were used concomitantly, each containing two of each of the sensor models mentioned above, at a total of eight sensors of each model. In each air quality monitor, the sensors were plugged in via USB to a Raspberry Pi, powered through Power Over Ethernet (PoE) and controlled using Python 3.6 libraries [35,36,37]. The data recorded by all the sensors were averaged over 10 s for cross-comparison purposes. Relative humidity and temperature were measured by each of the four air quality monitors using a Sensirion SHT35 [38] (±1.5% RH and ± 0.1 ${}^{\circ }C$).

The Plantower PMS5003 reports six size ranges called gr03um, gr05um, gr10um, gr25um, gr50um and gr100um, which represent, respectively, PNCs of particles >0.3
μm, >0.5
μm, >1 μm, >2.5
μm, >5 μm and >10 μm. The PNCs are reported as particles per 0.1 L of air. According to Sayahi et al. [39], the Plantower PMS5003 has a flow rate of ≈0.1
L/min and a wavelength of 640±10 nm with light polarised at $90^{\circ }$ [22]. The size ranges of the Plantower PMS5003 have been recalculated to obtain distinct size ranges similarly to Wallace et al. [14]. The size ranges obtained are as follows: n03_05, n05_10, n10_25, n25_50 and n50_100 which represent, respectively, PNCs of particles 0.3–0.5 μm, 0.5–1.0 μm, 1.0–2.5 μm, 2.5–5.0 μm and 5.0–10.0 μm.

The Sensirion SPS30 reports size ranges called n05, n1, n25, n4 and n10, which represent, respectively, PNCs in the range 0.3–0.5 μm, in the range 0.3–1 μm, in the range 0.3–2.5 μm, in the range 0.3–4 μm and in the range 0.3–10 μm. It utilises a laser beam of 660 nm wavelength and reports PNCs as $\mathrm{particles}/cm^{3}$. The Sensirion SPS30s are calibrated by their manufacturer against a TSI OPS 3330 or a TSI DustTrak DRX 8533. The accuracy of the calibration is then verified by the manufacturer using an atomized potassium chloride solution [40]. For the Sensirion SPS30, according to the manufacturer, particles above 4 μm are not directly measured but determined from the other size ranges using a particle distribution profile. The Sensirion SPS30 is certified for UK indicative monitoring and, although the sensors used in this study were acquired prior to the certification, private communication with the manufacturer confirmed that there had been no significant changes between the sensors used here and the sensors used for the certification. The size ranges of the Sensirion SPS30 were also recalculated to obtain distinct size ranges. The size ranges obtained are as follows: n03_05, n05_1, n1_25, n25_4 and n4_10 which represent, respectively, PNCs in the range 0.3–0.5 μm, in the range 0.5–1 μm, in the range 1–2.5 μm, in the range 2.5–4 μm and in the range 4–10 μm.

The Alphasense OPC-R1 is a single particle counter, which utilises a laser beam at a 639 nm wavelength, which can theoretically count up to 10,000 particles/s or 2500 particles/cm${}^{3}$ with a maximum coincidence probability of 0.7% at 1000 particles/cm${}^{3}$. The PNCs are output as $\mathrm{particles}/cm^{3}$ into 16 different size ranges (see Table 1) and can measure particles in the range 0.35–12.4 μm. It has a flow rate of 0.24 L/min. The Alphasense OPC-R1 was calibrated by its manufacturer using monodisperse Polystyrene Sphere (PLS) particles against an Alphasense OPC-R1, which itself had previously been calibrated against a TSI OPS 3330 [41].

### 2.2. Reference Instruments

#### 2.2.1. TSI OPS 3330

The TSI Optical Particle Sizer (OPS) 3330 (TSI Inc., Shoreview, MN, USA) is an optical instrument based on light scattering which reports PNCs divided into 16 size ranges (see Table 1), reported in $\mathrm{particles}/cm^{3}$, in the range 0.3–10 μm. It uses a laser beam at 660 nm and a flow rate of 1 L/min. It is calibrated by its manufacturer for size using Polystyrene
Sphere (PLS) [42]. In this study, the TSI OPS 3330 was used as a reference instrument and set to measure every 10 s. The size ranges of the TSI OPS 3330 were recalculated to match the size ranges of the different models of sensors to enable comparison. The method for redistributing the size ranges is described below in the Section 2.4.

#### 2.2.2. Aerasense Nanotracer

The Aerasense Nanotracer (Oxility BV, Venray, The Netherlands) counts particles in the range 10–300 nm based on diffusion charging. It is measuring below the advertised cut-off size of the low-cost sensors. It was calibrated by its manufacturer using KNO${}_{3}$ polydisperse particles and has a flow rate of 0.3–0.4 L/min. It is used because some of the sensors may be able to measure below 0.3 μm.

### 2.3. Experimental Conditions

The experimental conditions and set-up are the same as in Bulot et al. [30], the relevant elements are summarised here and the experimental set-up is described in Figure 2. The test chamber was placed in an environmental chamber where the temperature was controlled and set to 23 ${}^{\circ }C$. The temperature inside the test chamber varied in the range 26–29 ${}^{\circ }C$ throughout the experiments. Candle and incense smoke were used as the two different combustion sources, enabling testing at different particle size distributions. Candle smoke, here produced by smouldering, contains mostly particles in the range 0.02–0.1 μm with particle size peaks in the range 0.03–0.05 μm in terms of PNC [43,44]. For incense smoke, particles are mostly within the range 0.05–0.7 μm with a peak at 0.2 μm, in terms of PNC [45]. Five sets of experiments were conducted. For each set of experiments, several peaks of candle smoke were generated, followed by a longer concentration of candle smoke, then the air was cleaned of particles using the Electrostatic Precipitator (ESP), then a series of peaks of incense smoke were generated followed by a stable concentration of incense smoke. The peaks of PM lasted around 1 min and had a targeted concentration of 20–50 μg/m${}^{3}$ as measured in real-time by a DustTrak DRX 8533 Desktop (TSI Inc., Shoreview, RC, USA). During each experiment, the RH was set at different targets: 54, 69, 72, 76 and 79% RH. RH was controlled by a mist generator. RH was measured by the Sensirion SHT35 RH and temperature sensors built into each of the air quality monitors. RH was taken as the median reading of the four SHT35 sensors (as the distribution of RH recorded by the sensors was not normal). The experimental conditions are further described in Bulot et al. [30], along with some statistical analysis of the RH measured by the SHT35 sensors. Targeted RHs were set higher but could not be achieved with this experimental set-up. The range of RH attained in this study may not be sufficient to capture the impact of this environmental factor and the behaviour of the sensors may change for higher levels of RH.

The residual heat generated by the electronics and the sensors inside the air quality monitors means that the RH and temperature condition inside the air quality monitor are higher than the conditions surrounding the air quality monitors. For instance, in one of our previous studies, we noticed the RH was 15% lower inside the air quality monitor than outside [12]. It is not clear whether the particles have sufficient time to adjust to these conditions before being measured by the sensors. To avoid any impact of the phenomenon described above, the enclosure of the air quality monitors was removed during the experiment as can be seen in Figure 2.

The particle size distribution from the TSI OPS 3330 is available in Figure A13 and shows that, for candle-generated PM, there is less than 10 particles/cm${}^{3}$ above 5 μm and for incense-generated PM above 2.5 μm. Given the low values of PNC above 2.5 μm, only the size ranges of the sensors having a lower cut size < 2.5 μm will be considered during this study.

### 2.4. Data Analysis

#### 2.4.1. Feature Selection Methods

As detailed in the introduction of this paper, the outcome of feature selection methods depends on the methods chosen, and it is best practice to use different methods concomitantly and to compare their results. In this paper, we tested three methods: Ridge regression, Boruta and Recursive Feature Elimination (RFE) with Support Vector Machines (SVM). Ridge regression feature selection is an extension of the linear model with a penalisation term on the residual of the sums of squares using the L${}_{2}$ norm [46]. Ridge regression is an embedded method, as the selection of features is part of its algorithms. Boruta and RFE are both wrapper methods. Boruta is a wrapper based on random forest; it starts by adding shuffled copies of the existing variables to the dataset, called shadow variables. It then trains a random forest on this dataset and measures the variable importance (using the variable importance measure built into the random forest) and compares the importance of the initial variables to the importance of the shadow variables. Variables that obtained a significantly lower score than the higher score of the shadow variables are removed. It then reiterates the process [47]. RFE was initially developed to enable SVM to perform feature selection [48]. It trains a SVM model, computes a ranking criterion for all the variables considered (the weights of the SVM) and then removes the feature with the smallest ranking criterion. Filter-based methods have not been tested here as they generally do not allow for complex interactions between the variables considered. It is important to note that the scores obtained across the different methods cannot be compared; only the relative scores of each variable within a method can be compared.

In this study, the following variables are used for feature selection to predict each size range of each sensor: the source of PM (candle or incense), the number of particles <0.3
μm (measured by the Nanotracer), the number of particles in the range 0.3–0.8 μm (measured by the TSI OPS 3330), the number of particles in the range 0.3–10 μm (measured by the TSI OPS 3330) and RH. The feature selection methods are performed on the data from the five different levels of RH and using both sources of PM, aggregated by model of sensor.

#### 2.4.2. Lognormal Size Distribution

Particle size distribution is presented using lognormal distributions of the normalised concentrations calculated using the following formula, for each size range of the instrument considered:(1)$$\frac{dN}{dlog(D_{p})}=\frac{dN}{log\left(D_{p,u}\right)-log\left(D_{p,l}\right)}$$
with dN the PNC, $D_{p,u}$ the diameter of the upper boundary of the size range and $D_{p,l}$ the diameter of the lower boundary of the size range.

#### 2.4.3. Redistribution of OPS Size Ranges

The sensors and the TSI OPS 3330 measure the particle distribution using different numbers of size ranges or different cut-off diameters. In this study, we recalculate the size ranges of the TSI OPS 3330 to match the size ranges of each sensor tested. Overlapping size range fractions are computed with the formulas used by Di Antonio et al. [49] and shown in Figure 3 for a simple example. For this example, the equivalent TSI OPS 3330 size range $b_{eq}^{ops}$ is defined by:(2)$$b_{eq}^{ops}=b_{0}^{ops}\times f_{low}+b_{1}^{ops}+b_{2}^{ops}\times f_{upp}$$
with
(3)$$f_{low}=\frac{b_{0,upp}^{ops}-b_{low}}{b_{0,upp}^{ops}-b_{0,low}^{ops}}$$
and
(4)$$f_{upp}=\frac{b_{upp}-b_{2,low}^{ops}}{b_{2,upp}^{ops}-b_{2,low}^{ops}}$$

#### 2.4.4. Software and Data

The data were analysed using R 4.2.2 (R Foundation for Statistical Computing, Vienna, Austria) [50]. The underlying dataset is openly available at https://doi.org/10.5281/zenodo.7808620), and the code used for the analysis and to generate the tables and graphs of this study is openly available at https://doi.org/10.5281/zenodo.7808794. Boruta feature selection was conducted using the Boruta package [47]. RFE-SVM was conducted using the Caret package [51] using a 10 times repeated cross-validation, based on SVM radial [52]. Ridge was performed using the packages caret and glmnet [53] with a 10 times repeated cross-validation, and a grid search to optimise the penalisation coefficient λ between 0 and 1. The in-between variability between sensors of the same models was characterised by some of our previous works [12,30] which showed that this variability was relatively low. Therefore, the results presented here were aggregated over all the sensors for a given model, apart from the time series presented later for which a single sensor of each model was randomly selected. The size distributions of each individual sensor are presented in Appendix B.

## 3. Results

### 3.1. Correlation between the Different Size Ranges

Table 2, Table 3 and Table 4 present the correlations obtained by the different models of sensors across the different size ranges they report. For comparison, the same has been done for the equivalent size ranges calculated from the TSI OPS 3330 readings and these are presented between brackets in the same tables. This gives a baseline for the levels of correlation to expect in the actual size distribution of the particles measured. If the difference between correlation between the sensor size ranges and the correlation of the TSI OPS 3330 is >0.15, we consider that the two considered size ranges of the sensors are not truly independent. If the difference in correlation is <0.15, the size ranges of the sensors are considered independent.

For the Plantower PMS5003, the three size ranges were not independent from each other. For the Alphasense OPC-R1, all the size ranges were independent from each other. For the Sensirion SPS30, n03_05 and n05_1 were not independent from each other although to a lower extent than what was observed for the first size range of the Plantower PMS5003. n03_05 and n1_25 were independent from each other, and n05_1 and n1_25 were also independent from each other.

### 3.2. Time Series of the Experiments

The time series presented in Figure 4, Figure 5 and Figure 6 are focused on the experiment performed at 69% RH for brevity and the time series for the other experiments are available in Figure A1, Figure A2, Figure A3, Figure A4, Figure A5, Figure A6, Figure A7, Figure A8, Figure A9, Figure A10, Figure A11 and Figure A12. They show similar results to Experiment 2. The first seven peaks correspond to the generation of peaks of candle-generated PM followed by stable concentrations of candle-generated PM, then a series of six peaks of incense-generated PM and a stable concentration of incense-generated PM. For this section, the size ranges of the OPS have been converted to the size ranges of each individual sensor. The y-axis follows a logarithmic scale.

For the three models of sensors, the time series of the different size ranges closely followed the variation of the size ranges of the TSI OPS 3330, for both sources of particles. For the Plantower PMS5003, for candle-generated PM, the magnitude of sensor-reported PNC was 10 times lower than the magnitude of the TSI OPS 3330, for all the size ranges of this sensor model. For incense-generated PM, the first size range (0.3–0.5 μm) was also 10 times lower than the magnitude of the TSI OPS 3330, but the second size range (0.5–1 μm) obtained the same magnitude as the TSI OPS 3330 for peaks but underestimated PNC for the stable concentration of incense-generated PM. This size range also presented a downward slope for stable concentrations of incense-generated PM, which did not match the TSI OPS 3330 measurements. The third size range (1–2.5 μm) for incense-generated PM overestimated PNC for peak concentrations but, for the stable concentration, it started by over-reporting before then under-reporting. For the Sensirion SPS30, the magnitude was about 100 times lower for all the size ranges. For its first size range (0.3–0.5 μm) little difference was observed between candle- and incense-generated PM; for its second size range (0.5–1 μm), while the TSI OPS 3330 recorded lower PNC for incense-generated PM than for candle-generated PM, the Sensirion SPS30 recorded similar levels of PNC for the two sources. For the third size range (1–2.5 μm), the sensor under-reported more for incense than for candle-generated PM. For the Alphasense OPC-R1, the magnitude of the first size range (from 0.35–0.7 μm) was about 10 times lower; this size range also presented a lot of variability that was not present in the measurement made by the TSI OPS 3330. For the other size ranges of this sensor, the magnitude was similar to the TSI OPS 3330: for the size range 0.7–1.1 μm, the Alphasense OPC-R1 slightly over-reported PNC while, for the three remaining size ranges, it slightly under-reported PNC compared to the TSI OPS 3330.

### 3.3. Feature Selection

Table 5, Table 6 and Table 7 present the importance of the variables (source; PNC 0.01–0.3 μm; PNC 0.3–0.8 μm; PNC 0.3–10 μm; RH) for the different size ranges of the sensors, computed by using the three methods described in the Section 2.4.1: Boruta, Ridge and RFE-SVM. High scores denote the relevance of the variable to explain the size range considered. Each method computes their score differently and the values obtained should not be compared between the methods.

For the Plantower PMS5003, RH was consistently given a score of zero or close-to-zero. For the two first size ranges, the PNC for particles 0.01–0.3 μm was given the highest scores for the Boruta and Ridge method and a high score for the RFE-SVM method. The third size range was given lower scores for this variable for the three methods. The source of the PM was given relatively low scores for the first two size ranges and scores close to zero for the third size range. PNCs 0.3–0.8 μm and 0.3–10 μm were given high scores for all methods and all size ranges.

For the Sensirion SPS30, similarly, RH was given low scores for its three size ranges for all three methods. Source was given a low score for the first size range but obtained high scores for the second size range for Boruta and Ridge, and the highest scores for the third size range for Ridge and RFE-SVM. PNC for particles 0.01–0.3 μm was given relatively low scores on the three size ranges.

PNCs 0.3–0.8 μm and 0.3–10 μm were given the highest scores for all methods for the first size range and relatively high scores for the second size range. For the third size range, PNC 0.3–0.8 μm was given low scores for Ridge and RFE-SVM but a high score for Boruta, and PNC 0.3–10 μm was given high scores for Boruta and Ridge but a low score for RFE-SVM.

For the Alphasense OPC-R1, RH was given low scores, of zero or close-to-zero, for most size ranges and most methods except for Bin1 for Boruta and Ridge for which it was given moderate scores. Source was given the highest score for Bin2 to Bin4 for the three methods, the highest score for Bin1 for Boruta and Ridge, and a moderate score for RFE-SVM. The first size range was given lower scores for source. PNC for particles 0.01–0.3 μm was given relatively low scores for all methods for all size ranges. PNCs 0.3–0.8 μm and 0.3–10 μm were given moderate to high scores on all size ranges for the three methods considered.

### 3.4. Particle Size Distribution

Figure 7 presents the size distribution measured by the sensors and the TSI OPS 3330 during stable concentrations of candle- and incense-generated PM. The data are averaged per sensor model and are average over the five sets of experiments. The different size distributions recorded by the TSI OPS 3330 in each experiment are presented in Figure A13. To facilitate the visualisation, the data from the Sensirion SPS30 has been multiplied by 100 and the data from the Plantower PMS5003 by 10.

For the TSI OPS 3330, the incense-generated PM showed a relatively steeper decrease of PNC with increasing sizes and higher PNC with candle-generated PM for particles >0.5
μm than for incense-generated PM.

The Plantower PMS5003 reported almost the same distribution for candle- and incense-generated PM and surprisingly reported higher numbers for incense-generated PM than for candle-generated PM. For candle-generated PM, the magnitude was off by a factor 10. For incense-generated PM, the Plantower PMS5003 over-reported PNC for particles about <0.6
μm and under-reported for particles of wider diameter. The Sensirion SPS30 presented the same steeper decrease than the TSI OPS 3330 between candle- and incense-generated PM and detected fewer particles >0.7
μm for incense-generated PM than for candle-generated PM. As in the time series, the magnitude of this sensor differed by a factor of 100. For the Alphasense OPC-R1, the steeper decrease between incense- and candle-generated PM was also present. The magnitude of the first size range was much lower for the sensor than for the TSI OPS 3330 for both sources of PM but the other sizes followed each other with almost similar magnitude.

The readings of the individual sensors are available in Appendix B Figure A14, Figure A15 andFigure A16 and limited variability was observed between units of the same model of sensor for each size range although the Alphasense OPC-R1 demonstrated a higher variability for its first size range.

## 4. Discussion

The particle size distribution showed that Plantower PMS5003 did not capture the difference in size distribution between candle and incense smoke. Incense smoke had clearly fewer particles >0.5
μm than candle smoke, according to the TSI OPS 3330; however, while this was not captured by the Plantower PMS5003, it was captured by the Sensirion SPS30 to a certain extent and more clearly by the Alphasense OPC-R1. The size distribution captured by the latter sensor was close to the size distribution measured by the TSI OPS 3330, except for its first size range (0.35–0.7 μm). Ouimette et al. [22] found that the Plantower PMS5003 behaved as an integrating nephelometer, reporting correctly the aerosol scattering coefficient for particles in the range 0.26–0.46 μm. Using the global database of PurpleAir sensors, they also showed that the Plantower PMS5003 obtained similar shape of size distribution in different sites outdoors around the world, this being partly attributed to the fact that aerosol scattering coefficient of outdoor PM is generally constant. Similarly, He et al. [21] showed that the different size ranges of the Plantower PMS5003 had cut-off diameters in the range 0.1–0.7 μm, when testing the sensors with ammonium sulfate and sodium chloride. Together, these support the claim that the size distribution is computed by an algorithm rather than actually measured for each size ranges. Tryner et al. [19] studied the size segregation of the mass concentration reported by the Plantower PMS5003 and the Sensirion SPS30 in an environmental chamber, and exposed eight of each of these sensor models to stable concentrations of PM in the range 10–1000 μg/m${}^{3}$ lasting around 45 min each and generated using ammonium sulfate, Arizona road dust, NIST urban PM, wood smoke and oil mist with PLS of different diameters (0.1, 0.27, 0.72 and 2 μm). They found that the Plantower PMS5003 obtained similar shape of size distribution for all the diameters of PLS and for the different sources of pollution. However, Kuula et al. [23] exposed Plantower PMS5003 and Sensirion SPS30 sensors, amongst other sensor models, to monodisperse particles of diameters in the range 0.45–9.8 μm and they showed that, while the Plantower PMS5003 misclassified the size of the particles, it was still producing two different signals, one for particles 0.3–2.5 μm and one for particles 2.5–10 μm. Similarly, Zamora et al. [24] exposed the sensor to PLS of 0.081, 0.3, 0.8, 1.1, 2.5 and 4.8 μm and while the Plantower PMS5003 misclassified and misreported the size distribution of the particle measured, it showed some differences in the size distribution it reported between the different diameters tested. These suggest that the algorithm used by the Plantower PMS5003 may include a second measurement to compute the size distribution it reports. This is further supported by: (1) the between size range correlations obtained here by the sensor; and (2) the Plantower PMS5003 differences between the scores obtained for PNC <0.3
μm for its first two size ranges and the third size range.

The Sensirion SPS30 captured some of the variations in the size distribution between incense and candle-generated PM. Tryner et al. [19] obtained two different sizes, using sensor-reported PM mass concentrations, for PLS particles of 0.1 and 0.27 μm, and of 0.72 and 2 μm, in contrast to the Plantower PMS5003. Nonetheless, in their study, the Sensirion SPS30 did not agree with the Aerodynamic Particle Sizer Spectrometer that was used as a reference instrument. Kuula et al. [23], again using sensor-reported PM mass concentrations, suggested that this sensor was able to differentiate two different size ranges, 0.3–0.9 μm and 0.7–1.3 μm, with a valid detection range for PM${}_{1}$ mass concentration. The correlations obtained here between the different size ranges of the sensors support the fact that this sensor is able to differentiate two size ranges: 0.3–0.5 μm and 1–2.5 μm.

The Alphasense OPC-R1 showed clear differences between its size ranges, which generally followed the correlation obtained by the TSI OPS 3330, for both PM sources. In our 2020 study [30], which analysed the same set of experiments but focused on sensor-reported mass concentrations, the Alphasense OPC-R1 obtained lower correlation coefficients between the mass concentration of the sensors and a DustTrak DRX 8533 for incense- than for candle-generated PM. This could have several explanations: (1) the algorithm used by the Alphasense OPC-R1 to convert PNC to mass concentration is based on factors, which differ according to specific properties of the particles measured; (2) there are some differences in the measurement taken by the TSI OPS 3330 and the DustTrak DRX 8533, which is unlikely as they are based on the same technology and are made by the same manufacturer.

For the three models of sensors, the variables that impacted their readings the most were the variables linked to the particle size distribution, followed by source, with RH having less impact on the readings. This corroborates the suggestion in our earlier paper [30] that the performances of the sensors are primarily impacted by the size distribution of the particles and secondarily the source of those particles. This also means that, at RH < 79%, this variable does not need to be corrected. This would need to be verified on different sources of PM, especially with sources having a different size distribution and refractive index than candle and incense. Jayaratne et al. [54] showed that the Plantower PMS1003 and a DustTrak DRX8530 started to over-report PM mass concentration for RH > 75%. Tryner et al. [19] also found an impact of RH on the readings of the Plantower PMS5003 and the Sensirion SPS30 for RH > 75–80%. Conversely, Holder et al. [55] and Liang et al. [56], who both studied the response of the sensors to wild fire smoke, showed that it was not necessary to include RH in the correction models. Similarly, in a year-long outdoor study in a port city, Bulot et al. [12] showed only marginal improvements in models including RH compared to models not including it. Taken together, these suggest that the impact of RH is highly dependent on the composition of the aerosol being measured. This would explain why different studies conducted in different settings, in different places or countries, obtain conflicting results with regard to RH.

We found here that the accuracy of the readings of the first two size ranges of the Plantower PMS5003 were impacted by PNC <0.3
μm. This is similar to the results obtained by both He et al. [21], who developed a transfer-function based model that predicted that the sensor would output a signal for particles with diameter <0.3
μm, and by Ouimette et al. [22] whose physical model of the Plantower PMS5003 as an integrating nephelometer, based on the Mie theory, also predicted that the sensor would be able to measure particles <0.3
μm, in direct proportion to their contribution to the aerosol scattering coefficient. The Alphasense OPC-R1 and the Sensirion SPS30 readings were not impacted by the PNC in that size range. While it is quite clear that the Alphasense OPC-R1 is an Optical Particle Counter (OPC), this, along with the differences observed earlier on the sensitivity to particle size distribution between the Plantower PMS5003 and the Sensirion SPS30, may suggest that the Sensirion SPS30 is not an integrated nephelometer and/or measures and interprets the PNC differently from the Plantower PMS5003. Tryner et al. [19] also suggested that these two sensors had a different method for measuring or interpreting light-scattering data.

This experimental set-up has some limitations that must be accounted for before extrapolating its results. The sensors were only tested against two combustion sources of pollution, and environmental particles will have different physical and chemical properties (i.e., hygroscopicity, refractive index and particle size). Additionally, the range of RH was limited and the temperature was set in the range 26–29 ${}^{\circ }C$ which does not reflect the range of environmental conditions to which the sensors would be exposed in outdoor conditions.

As these sensors are expected to be used in large monitoring networks, for extended periods of time, it is critical to characterise their potential drift with time. This is not possible with the data collected during this study. There is conflicting evidence in the literature on this subject. Tryner et al. [19], in their laboratory experiment, simulated the ageing of the Sensirion SPS30 and the Plantower PMS5003 by exposing them to mass concentrations in the range 7300–33,000 μg/m${}^{3}$ for 18 h to emulate the outdoor concentrations these sensors would encounter over a year, and showed that some performance degradation occurred for three out of eight of the Plantower PMS5003s tested but not for the Sensirion SPS30. However, it is likely that the really high concentrations to which the sensors were exposed may over-estimate their time drift. Indeed, in our previous outdoor study [57], the performances of different low-cost sensors, including the Plantower PMS5003, were evaluated over a year outdoor, and no performance degradation was found. Several studies lasting from a couple of months to a year or more found no drift over time of the sensors [24,58,59,60]. Wallace et al. [16] studied eight Purple Air sensors for 1.5 to 3 years indoors and outdoors, and found little evidence to support a temporal drift. Similarly, Collier-Oxandale et al. [61] conducted a three year long analysis of the performances of a network of 400 Purple Air II sensors (which use two Plantower PMS5003 each) deployed across 14 communities in the USA, and found that the performances of the sensors were mainly explained by seasonal variations but found little evidence of a temporal drift.

Although the three models of sensors were able to capture the temporal variations of the PNC, as reported by the TSI OPS 3330, the only sensor that reliably reported the particle size distribution of the aerosols was the Alphasense OPC-R1.

## 5. Conclusions

In this study, eight sensors of each of three models, Alphasense OPC-R1, Plantower PMS5003 and Sensirion SPS30, at a total of 24 sensors, were studied at a 10 s resolution and exposed to short-lived events of PM pollution, generated from two combustion sources having a different size distribution profile, at varying levels of RH.

The time series obtained revealed that the sensors were able to closely follow PNC variation measured by the reference TSI OPS 3330, for both sources of PM, but for the Plantower PMS5003 and the Sensirion SPS30 PNC measurements recorded were, respectively, 10 and 100 times lower than the measurements of the TSI OPS 3330. For the Sensirion SPS30, the second and third size ranges under-reported more than the first size range for incense-generated PM than for candle-generated PM. The magnitude was correct for the Alphasense OPC-R1, except for its first size range, 0.35–0.7 μm.

Regarding the independence of the size range reported by the sensors and their accuracy, the Plantower PMS5003 reported two independent signals; the Alphasense OPC-R1 reports an independent signal for each of its size ranges; the Sensirion SPS30 reported two independent signals for 0.3–0.5 μm and 1–2.5 μm. The analysis conducted suggested that the Plantower PMS5003 and the Sensirion SPS30 had a different method for measuring or interpreting light-scattering data, and reporting the determined PNC.

For capturing the particle size distribution, the Plantower PMS5003 showed no difference between incense- and candle-generated PM, while the two other sensors recorded differences that were also recorded by the TSI OPS 3330. The Alphasense OPC-R1 measured values that were close to the data reported by the TSI OPS 3330, except for its first size range (0.35–0.7 μm).

The analysis of the feature selection revealed that the sensors were more susceptible to the composition of the particles and their size distribution than to RH at the levels of humidity considered. We therefore recommend that a RH correction is not required below 75–79%. PNC in the range 0.01–0.3 μm impacted the first size range of the Plantower PMS5003 supporting the fact that this sensor is an integrating nephelometer.

For studies requiring more detailed knowledge of the particle size distribution of the aerosol measured, the Alphasense OPC-R1 should be preferred, although our previous study [30] also showed that this sensor was less suited to report PM mass concentration for the two sources of PM used here. If a more general image of the particle size distribution is sufficient, the Sensirion SPS30 should be considered. It is not clear from the results of this study whether the Plantower PMS5003 can be used in this scenario but it can be used to measure the general trends of PNC. This work shows that there is added value in directly using the PNC instead of PM mass concentration, as the size ranges provide some level of information about the particle size distribution, especially in the case of the Alphasense OPC-R1. This differential information collected by the PNC size ranges can be used to improve the calibration models developed to calibrate the sensors to standard performances and provide extra granularity with regard to source profiling.

## Figures and Tables

**Figure 1 sensors-23-07657-f001:**
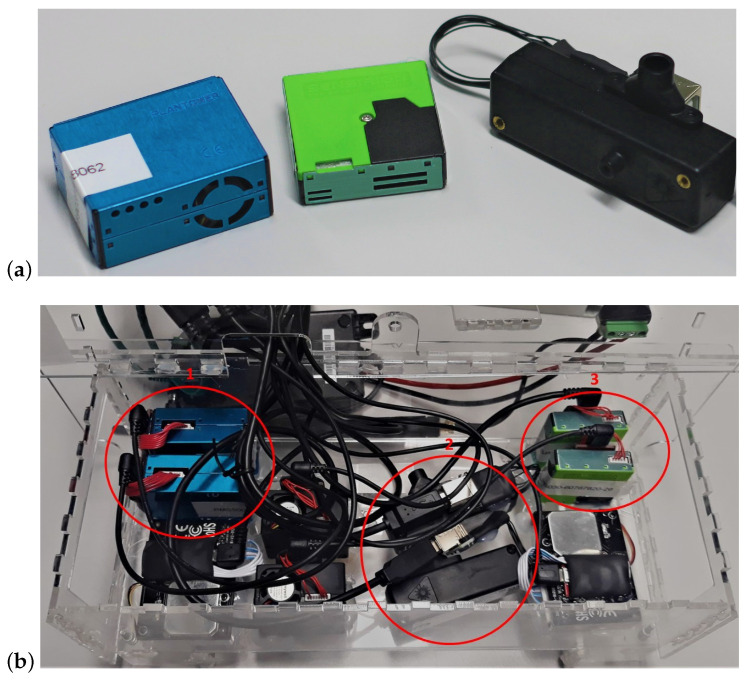
(**a**) Sensors tested from left to right: Plantower PMS5003, Sensirion SPS030, Alphasense OPC-R1, adapted from Bulot et al. [30]. (**b**) Position of the sensors tested within each air quality monitor. From left to right, top to bottom: two Plantower PMS5003s (red circle 1), one Novafitness SDS018, two Honeywell HPMA115S0s, two Alphasense OPC-R1 (red circle 2), two Sensirion SPS30s (red cicle 3) and one Novafitness SDS018. All the inlets are facing down. Reprinted/adapted with permission from Bulot et al. [30]. 2020, by the authors.

**Figure 2 sensors-23-07657-f002:**
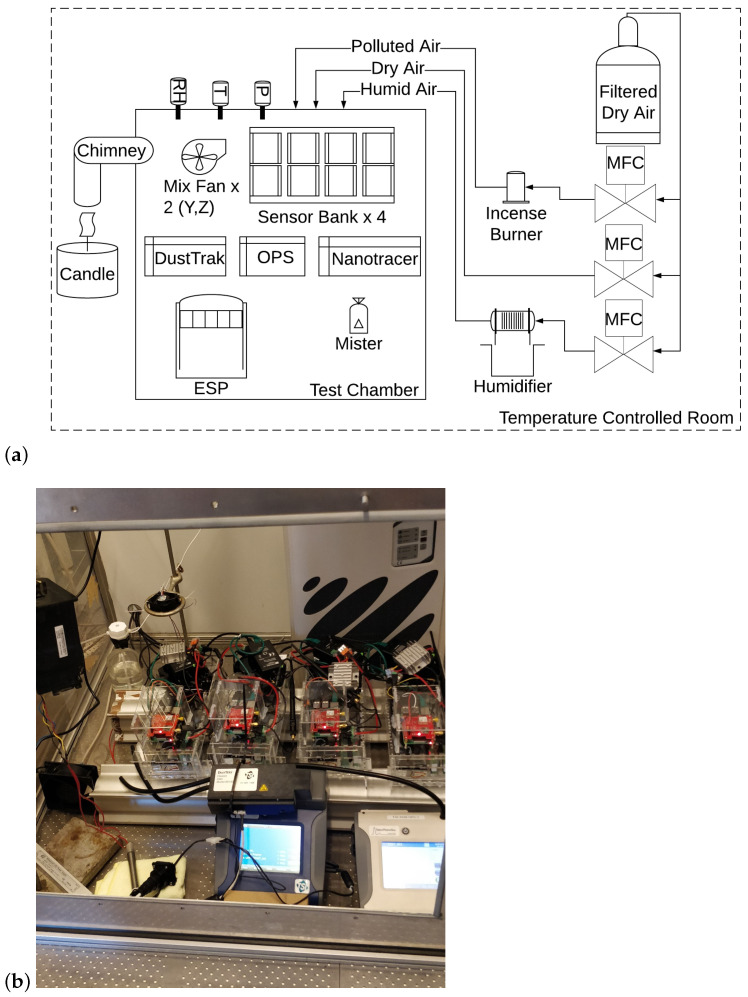
(**a**) Schematic showing the arrangement of the test chamber and supporting equipment, adapted from Bulot et al. [30]. (**b**) Image showing the air quality boxes located in the test chamber. Reprinted/adapted with permission from Bulot et al. [30]. 2020, by the authors. Mass Flow Controllers (MFCs).

**Figure 3 sensors-23-07657-f003:**
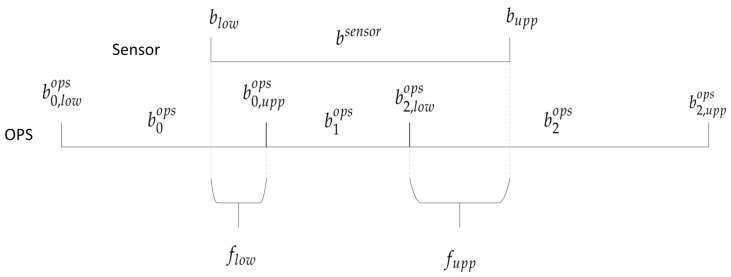
Principle of the redistribution of the size ranges of the TSI OPS 3330 to match the size ranges of each model of sensor with $b^{sensor}$ the sensor size range, $b_{low}$ and $b_{upp}$ the lower and upper cut size of the sensor size range, $b_{i}^{ops}$ the *i* corresponding size ranges of the TSI OPS 3330, $b_{i,low}^{ops}$ and $b_{i,upp}^{ops}$ the lower and upper cut sizes of the $i^{th}$ size range of the TSI OPS 3330, and $f_{low}$ and $f_{upp}$ the lower and upper fractions of the size ranges corresponding to $b_{low}$ and $b_{upp}$.

**Figure 4 sensors-23-07657-f004:**
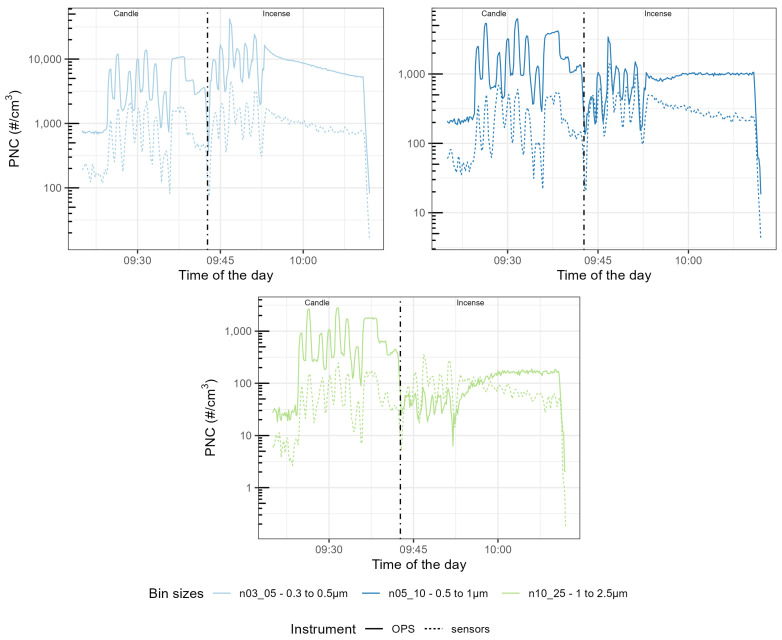
Time series of the Particle Number Concentration (PNC) size ranges of the Plantower PMS5003 compared with the size ranges computed from the OPS size ranges. On the left of the dotted line, PM was generated using a candle, on the right using incense. The different categories are, from left to right and top to bottom, n03_05 (in the range 0.3–0.5 μm), n05_10um (in the range 0.5–1 μm), and n10_25um (in the range 1–2.5 μm).

**Figure 5 sensors-23-07657-f005:**
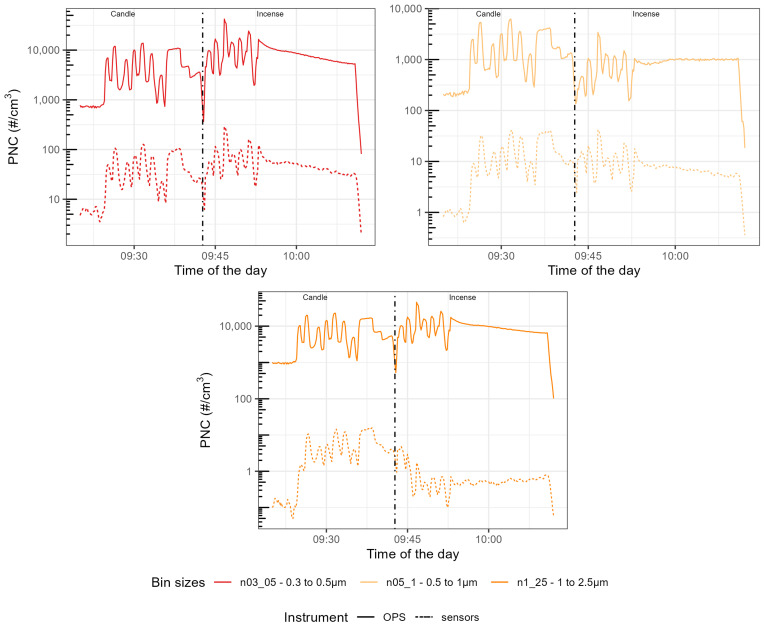
Time series of the Particle Number Concentration (PNC) size ranges of the Sensirion SPS30 compared with the size ranges computed from the OPS size ranges. On the left of the dotted line, PM was generated using a candle, on the right using incense. The different categories are, from left to right and top to bottom, n03_05 (in the range 0.3–0.5 μm), n05_1 (in the range 0.5–1 μm), and n1_25 (in the range 1–2.5 μm).

**Figure 6 sensors-23-07657-f006:**
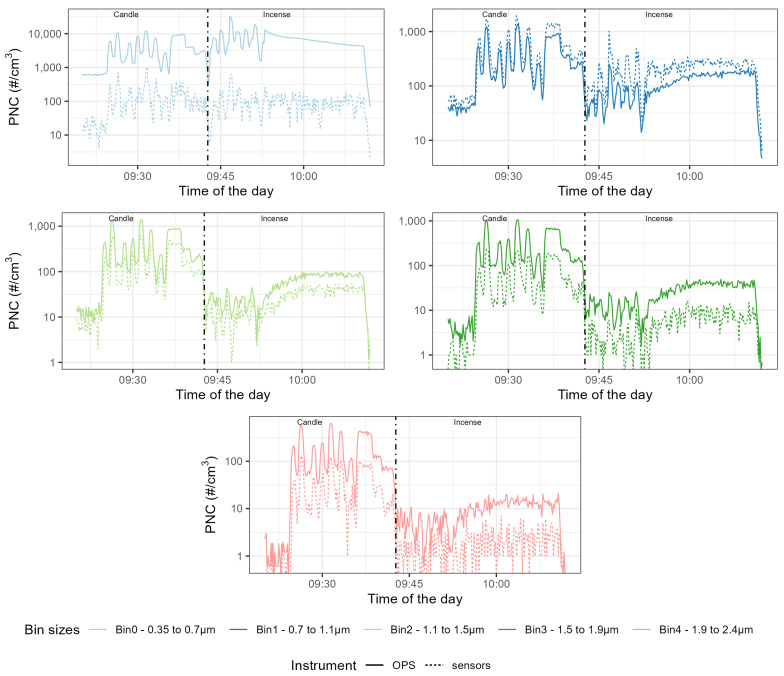
Time series of the Particle Number Concentration (PNC) size ranges of the Alphasense OPC-R1 compared with the size ranges computed from the OPS size ranges. On the left of the dotted line, PM was generated using a candle, on the right using incense. The different categories are, from left to right and top to bottom, Bin0 (in the range 0.4–0.7 μm), Bin1 (in the range 0.7–1.1 μm), Bin2 (in the range 1.1–1.5 μm), Bin3 (in the range 1.5–1.9 μm) and Bin4 (in the range 1.9–2.4 μm).

**Figure 7 sensors-23-07657-f007:**
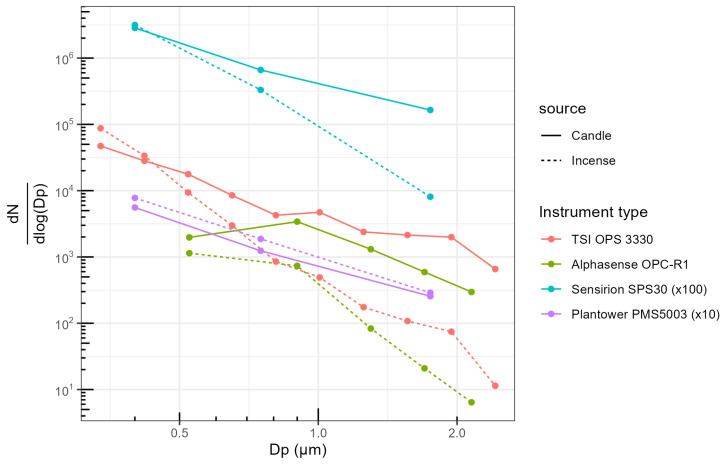
Particle size distribution reported by the sensors and the OPS TSI for stable concentrations of incense- and candle-generated PM, aggregated over the five experiments and per sensor model. For each sensor model, the data presented is the average per size range. The dots are on the midpoint of the size ranges. To facilitate the visualisation, the data from the Sensirion SPS30 has been multiplied by 100 and the data from the Plantower PMS5003 by 10. The dots represent the actual datapoints used to construct the plot.

**Table 1 sensors-23-07657-t001:** Size ranges of the Plantower PMS5003, Sensirion SPS30, Alphasense OPC-R1 and TSI OPS 3330.

Sensor	Size Ranges (μm)
PMS5003	>0.3; >0.5; >1; >2.5; >5; >10
SPS30	0.3–0.5; 0.3–1; 0.3–2.5; 0.3–4; 0.3–10
OPC-R1	0.4–0.7; 0.7–1.1; 1.1–1.5; 1.5–1.9; 1.9–2.4;
	2.4–3; 3–4; 4–5; 5–6; 6–7; 7–8;
	8–9; 9–10; 10–11; 11–12; 12–12.4
OPS 3330	0.3–0.4; 0.4–0.5; 0.5–0.6; 0.6–0.7; 0.7–0.9; 0.9–1.1;
	1.1–1.4; 1.4–1.7; 1.7–2.2; 2.2–2.7; 2.7–3.3;
	3.3–4.2; 4.2–5.2; 5.2–6.5; 6.5–8.0; 8.0–10

**Table 2 sensors-23-07657-t002:** Correlation and linear model between the different Particle Number Concentration (PNC) size ranges reported by the Plantower PMS5003 during the period of the study. The numbers between brackets represent the correlation between equivalent size ranges of the TSI OPS 3330. Cells are shaded in red if the difference between the correlation of the sensor and the TSI OPS 3330 is greater than 0.15. If the sensor size ranges accurately measured the size distribution, they should obtain a similar correlation to the TSI OPS 3330.

PMS5003			n05_10	n10_25
		Range	0.5–1 μm	1–2.5 μm
$R^{2}$	n03_05	0.3–0.5 μm	0.99 (0.53)	0.86 (0.14)
$R^{2}$	n05_10	0.5–1 μm		0.80(0.64)

**Table 3 sensors-23-07657-t003:** Correlation and linear model between the different Particle Number Concentration (PNC) size ranges reported by the Sensirion SPS30 during the period of the study. The numbers between brackets represent the correlation between equivalent size ranges of the TSI OPS 3330. Cells are shaded in red if the difference between the correlation of the sensor and the TSI OPS 3330 is greater than 0.15. If the sensor size ranges accurately measured the size distribution, they should obtain a similar correlation to the TSI OPS 3330.

SPS30			n05_1	n1_25
		Range	0.5–1 μm	1–2.5 μm
$R^{2}$	n03_05	0.3–0.5 μm	0.73(0.53)	0.18 (0.14)
$R^{2}$	n05_1	0.5–1 μm		0.70 (0.64)

**Table 4 sensors-23-07657-t004:** Correlation and linear model between the different Particle Number Concentration (PNC) size ranges reported by the Alphasense OPC-R1 during the period of the study. The numbers between brackets represent the correlation between equivalent size ranges of the TSI OPS 3330. Cells are shaded in red if the difference between the correlation of the sensor and the TSI OPS 3330 is greater than 0.15. If the sensor size ranges accurately measured the size distribution, they should obtain a similar correlation to the TSI OPS 3330.

OPCR1			Bin1	Bin2	Bin3	Bin4
		Range	0.7–1.1 μm	1.1–1.5 μm	1.5–1.9 μm	1.9–2.4 μm
$R^{2}$	Bin0	0.35–0.7 μm	0.31 (0.43)	0.19 (0.17)	0.14 (0.09)	0.12 (0.06)
$R^{2}$	Bin1	0.7–1.1 μm		0.78 (0.80)	0.61 (0.64)	0.53 (0.53)
$R^{2}$	Bin2	1.1–1.5 μm			0.93 (0.96)	0.88 (0.89)
$R^{2}$	Bin3	1.5–1.9 μm				0.96 (0.98)

**Table 5 sensors-23-07657-t005:** Scores of the feature selection methods for the different size ranges of the Plantower PMS5003 computed using (a) Boruta, (b) Ridge, (c) RFE-SVM. High scores denote the relevance of the variable to explain the size range. Each method computes the score differently and the values obtained should not be compared between the methods. PNC 0.01–0.3 μm is measured by the Nanotracer and the other two size fractions are measured by the TSI OPS 3330.

PMS5003	n03_05			n05_1			n10_25		
	0.3–0.5 μm			0.5–1 μm			1–2.5 μm		
Method	(a)	(b)	(c)	(a)	(b)	(c)	(a)	(b)	(c)
Source	18	31	13	22	38	17	6	1	0
PNC 0.01–0.3 μm	49	100	50	52	100	52	16	30	24
PNC 0.3–0.8 μm	29	75	63	27	67	59	30	95	80
PNC 0.3–10 μm	28	65	60	26	51	55	33	100	81
RH	9	0	0	10	0	0	7	0	0

**Table 6 sensors-23-07657-t006:** Scores of the feature selection methods for the different size ranges of the Sensirion SPS30 computed using (a) Boruta, (b) Ridge, (c) RFE-SVM. High scores denote the relevance of the variable to explain the size range. Each method computes the score differently and the values obtained should not be compared between the methods.

SPS30	n03_05			n05_1			n1_25		
	0.3–0.5 μm			0.5–1 μm			1–2.5 μm		
Method	(a)	(b)	(c)	(a)	(b)	(c)	(a)	(b)	(c)
Source	6	0	4	32	68	5	6	100	23
PNC 0.01–0.3 μm	17	26	22	16	0	6	17	0	5
PNC 0.3–0.8 μm	31	100	79	26	36	35	31	11	3
PNC 0.3–10 μm	33	84	79	27	100	40	33	76	5
RH	9	8	0	16	28	2	7	16	3

**Table 7 sensors-23-07657-t007:** Scores of the feature selection methods for the different size ranges of the Alphasense OPC-R1 computed using (a) Boruta, (b) Ridge, (c) RFE-SVM. High scores denote the relevance of the variable to explain the size range. Each method computes the score differently and the values obtained should not be compared between the methods.

OPC-R1	Bin0			Bin1			Bin2			Bin3			Bin4		
	0.35–0.7 μm			0.7–1.1 μm			1.1–1.5 μm			1.5–1.9 μm			1.9–2.4 μm		
Method	(a)	(b)	(c)	(a)	(b)	(c)	(a)	(b)	(c)	(a)	(b)	(c)	(a)	(b)	(c)
Source	17	22	10	35	100	16	74	100	35	69	100	36	69	100	35
PNC 0.01–0.3 μm	12	3	23	10	1	3	14	15	5	13	12	5	13	10	5
PNC 0.3–0.8μm	35	100	98	20	0	32	23	33	15	23	34	13	21	35	13
PNC 0.3–10 μm	29	85	96	21	81	34	33	95	16	32	95	15	32	96	14
RH	5	0	0	15	46	7	8	0	0	5	0	0	4	0	0

## Data Availability

The underlying dataset is openly available at https://doi.org/10.5281/zenodo.7808620, and the code used for the analysis and to generate the tables and graphs of this study is openly available at https://doi.org/10.5281/zenodo.7808794.

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
