# Peer review of "Laboratory Comparison of Low-Cost Particulate Matter Sensors to Measure Transient Events of Pollution—Part B—Particle Number Concentrations"

_sensors, 2023, doi:10.3390/s23177657_

Round 1
Reviewer 1 Report
This paper is well written and presents interesting information on the users of the three instruments tested.
All of them are claimed to provide the number of particles according to a certain size range (the authors call them size bins, but probably the term size ranges should be preferred).
The experimental set-up and the comparison with a reference Optical Particle Counting is considered satisfactory, even though the particles generated by the two combustion sources are not really reflecting the size distribution found in ambient atmosphere.
In fact, Authors consider low sensors deployed in high number, as the primary tools for a better description of atmospheric pollution in a given zone.
However, some attention should be given to the following aspects:
· Metrics used for particulate matter are mostly based on mass concentration. Several decades ago, standards were elaborated in terms of total suspended particulate matter (TSP), then as PM10 and, later, as PM2,5. These standards were promoted according to health effects. The size distribution could be a promising metric; however, it cannot be easily directly related to mass concentration. This aspect should be added in the paper preface.
· Since the sensors are expected to be used in large monitoring sites, the paper should report information and data (if available) about the time stability of the instruments; at least in terms of the initial factory calibration. This is because the response of optically based instruments drifts with time. The experiments reported by the Authors lasted for a short time, compared with the time evolution of pollution episodes that is of the order of hours or days and even more. It is suggested to present some available results to better inform the individuals involved in air pollution assessment on this aspect.
· The scattering efficiency for particles below 0.3µm is very low, thus the responses in the lower size ranges is strongly affected by several parameters impacting the final relationship between geometric size and optical size as measured by the instruments.
·
In conclusion, the paper is worth publishing taking into a proper account the above considerations.
Reviewer 3 Report
Formulas (3) and (4) for size bin fractions flow and fupp are wrong. As a result, all calculations in which they were used are flawed.
The procedure for calculating the PNC in the equivalent TSI OPS 3330 size bin is not described.
Must revise according to reference [44]
Reference [42] and [46] is duplicated.
Reviewer 4 Report
Comments for authors
Abstract
1) It needs to contain some results obtained in the article, in order to arouse the reader's interest.
2) “Particle detectors generally focus on the mass particulate matter concentrations reported by these sensors or report particle number (PNC) concentrations. In the study presented here it was on Type sensors (PNC)”.
I suggest that in the article make a differentiation of how these two types of sensors work, in order to show their differences. This is necessary because at the beginning of the introduction there is a sentence “Given the recently substantially reduced WHO exposure limits, up to 5 µg/m3”
See that 5 µg/m3” and unit of concentration (mass).
3) In general, this summary can be rewritten to improve your understanding of the objectives of the article.
Introduction
4)The introduction also does not contain any results.
5) There are paragraphs in the introduction that could be in the methodology, for example the description of the sensors used in the experiment.
6) Because they used a simulation in the experiment (candle and incense burning), they could have used a source of real particulates, industries and combustion of diesel vehicles, etc..
7)Objectives can this introduction, but not at the end usually the end and to present some results obtained.
8) I also did not see in the discussions and conclusion if all the research objectives were achieved
Materials and methods
9) It was not placed in the description of the equipment, which type of laser and model it is, it has only the wavelength.
10) I found the description of the methodology (of the experiment) confusing, this can be improved
11) Parts of figures 4, 5 and 6 are low quality, difficult to visualize. Figures A1, A3 A4 A6 A7 A8 A9 A10 also have the same error.
12) In the data analysis, the possible statistical errors of the measurements are not mentioned, since 8 instruments of each type were used, how to compare these results with each other.
13) I suggest better describing the results section it is confusing for the reader's understanding
14) I suggest rewriting the conclusion, pointing out the most important results of valuing the results obtained
15) The suggestions presented here aim to improve the quality of the description of the article (understanding), there are parts that confuse the writing
Comments
Minor editing of English language required
Round 2
Reviewer 2 Report
The authors have admirably responded to earlier comments and have redone various calculations with the result of measurably improving their paper.
I have just two comments in the attached Word "track changes" file.

Reviewer 3 Report
No comments or suggesstions
Author Response
Many thanks.